# Bio-Mercury Remediation Suitability Index: A Novel Proposal That Compiles the PGPR Features of Bacterial Strains and Its Potential Use in Phytoremediation

**DOI:** 10.3390/ijerph18084213

**Published:** 2021-04-16

**Authors:** Marina Robas, Pedro A. Jiménez, Daniel González, Agustín Probanza

**Affiliations:** Department of Pharmaceutical Science and Health, Montepríncipe Campus, CEU San Pablo University, Ctra. Boadilla del Monte Km 5.300, 28668 Boadilla del Monte, Spain; pedro.jimenezgomez@ceu.es (P.A.J.); daniel.gonzalezreguero@ceu.es (D.G.); a.probanza@ceu.es (A.P.)

**Keywords:** heavy metal pollution, bioremediation, PGPR, BMR-SI

## Abstract

Soil pollution from heavy metals, especially mercury, is an environmental problem for human health. Biological approaches offer interesting tools, which necessarily involve the selection of organisms capable of transforming the environment via bioremediation. To evaluate the potential use of microorganisms in phytorhizoremediation, bacterial strains were isolated from rhizospheric and bulk soil under conditions of chronic natural mercury, which were identified and characterized by studying the following: (i) their plant growth promoting rhizobacteria (PGPR) activities; and (ii) their maximum bactericide concentration of mercury. Information regarding auxin production, phosphate solubilization, siderophore synthesis and 1-aminocyclopropane-1-carboxylic acid deaminase (ACCd) capacity of the isolates was compiled in order to select the strains that fit potential biotechnological use. To achieve this objective, the present work proposes the Bio-Mercury Remediation Suitability Index (BMR-SI), which reflects the integral behavior of the strains for heavy metal polluted soil bioremediation. Only those strains that rigorously fulfilled all of the established criteria were selected for further assays.

## 1. Introduction

Mercury (Hg) is an element with a high level of toxicity and is a serious environmental problem, which can ultimately be transmitted to the food chain, and consequently affect human health [1]. This can be explained because of its high potential for bioaccumulation and biomagnification, due to the solubility of Hg and methylmercury in fat tissue and muscle. From a toxicological point of view, it is a toxic metal; this means that it does not have a specific biological role, which means that its incorporation into the body is not necessary. Therefore, from certain doses it produces adverse health effects, such as problems in the development, growth, and reproduction of living beings [2]. 

Hg anthropogenic soil contamination began with the industrial revolution and has now become a matter of global, economic, public health, and environmental conservation concern. The Almadén (Ciudad Real) mining area has been exploited over the years and is considered one of the largest Hg production areas in the world. Mineral Hg deposits are found predominantly as cinnabar (HgS). Elemental mercury (Hg^0^) is also present and locally abundant in the atmosphere, where it can stay up to 1.7 years [3]. Given the large size of Almadén, Hg dispersion by rivers and emissions throughout the 2000 years that the mining district has been active, it is considered one of the most contaminated areas of the planet, both due to the natural origin of Hg, and due to its anthropogenic exploitation. When the exploited mine closed, other forms of land use had to be considered, such as agriculture or livestock. 

There are physicochemical methods that allow for the removal of this metal from the soil, but currently there is a trend toward the use of biological lines of action that are more respectful of the environment, based on the use of biotechnological methods such as bioremediation. This is the case of phytorhizoremediation based on the use of bacteria capable of participating in the recycling of elements and chemical compounds in nature [4]. It has provided good results in reducing heavy metal contamination in soils, including those contaminated with Hg [5], since rhizobia and other rhizospheric bacteria can increase the bioavailability of heavy metal ions and favor their absorption by the plant [6]. An example are plant growth promoting rhizobacteria (PGPR) [7], which can be used as partners in phytorhizoremediation both by helping the plant physiologically in its development to remove the metal, as well as through its direct effect on the contaminant.

The selective pressure exerted by environments contaminated with this type of toxic metal has allowed for the development of microbial resistance systems [8,9]. Numerous studies have been carried out on the characterization of bacterial communities, their response to contaminating agents, as well as the sequencing and characterization of the genes responsible for the transformation of these compounds [10]. The determinants of Hg resistance have been found in a wide range of bacteria and are based on genes clustered in the mer operon [11]. The presence of a contaminant favors the selection of naturally resistant strains and of strains capable of acquiring resistance mechanisms from mobile gene elements such as integrons, transposons, or plasmids [12].

The selection of this type of gene may involve the co-selection of other genes that modify the phenotype of the recipient bacteria through cross-selection phenomena [13], described for antibioresistance together with resistance to Hg [14]. In fact, the dispersion of genes with bacterial resistance to antibiotics have been considered environmental pollutants whose prevalence increases as a result of certain human activities [15]. In the particular case of bacteria capable of living under the pressure of Hg contamination, the description offered has been that together with contributing to the emergence of resistance that promotes the survival of microorganisms, it can also lead to increased plasmid resistance to antibiotics [14].

Given that mercury-tolerant and mercury-resistant microorganisms can contribute to reducing or eliminating the contamination of different forms of Hg in the environment, there is growing interest in the selection of strains with potential biotechnological and bioremediative use [16], although it is equally necessary to avoid the dispersal of microbial genes that may contribute to increasing other forms of biological contamination.

In subsequent tests on the selected strains, their suitability has been evaluated from a broader point of view in order to ensure their safety when reintroduced into the environment through bioremediation, for example, from the point of view of their antibiotic resistant profile. The objective is to avoid turning an environmental pollution problem into a potential public health problem. The strains used in this type of process must guarantee safety at all levels; both for humans and for the natural environment (animals and plants).

## 2. Materials and Methods

### 2.1. Obtaining the Mercury-Tolerant Strains

This study was carried out with samples obtained in the mining district of Almadén, Ciudad Real (Spain). The experimental plot (Plot M) was used, which was classified as an area of high contamination with 1710 mg/kg Hg [17]. The plants sampled were *Rumex induratus* Boiss. & Reut. (A), *Rumex bucephalophorus* L. (B), *Avena sativa* L. (C), *Medicago sativa* L. (D), and *Vicia benghalensis* L. (E).

For the extraction of bacterial communities present in the rhizosphere and bulk soil, the modified method described by García-Villaraco et al. [18] was followed. To do so, 2 g of soil were re-suspended in 20 mL of sterile saline solution (NaCl 0.45%). The mixture was homogenized using blades at 16,000 r.p.m., for 2 min. Afterward, it was centrifuged at 2500 r.p.m. for 10 min.

For the isolation of bacteria resistant to Hg, 1 mL of the supernatant (microbial extraction) was seeded in masse on standard method agar plates (SMA, Pronadisa^®^, Madrid, Spain) supplemented with different concentrations of Hg in its salt form HgCl_2_: 400, 320, 200, 160, 100, 80, 50, 40, 25, 20, 10, and 0 µg/mL. The sown plates were incubated for 24 h at 37 °C in aerobiosis. Bacteria resistant to Hg were considered to be those that met the criteria of [19] Mathema et al.; those that were Gram-positive with a maximum actericidal concentration (MBC) of 40 µg/mL and those that were Gram-negative with 30 µg/mL. Growth was found up to concentrations of 160 µg/mL of HgCl_2_.

Using these criteria, a total of 149 strains were isolated. In order to obtain pure cultures, they were re-isolated in SMA plates supplemented with 30 µg/mL of Hg.

### 2.2. Identification of the Bacterial Isolates

Morphological (Gram staining) and biochemical characterization of the bacterial isolates was carried out for preliminary identification. Molecular Identification was conducted using 16s rDNA gene sequencing of the V3-V4 region using the primers rP2 (5′ ACGGCTACCTTGTTACGACTT 3′) and fD1 (5′ AGAGTTTGATCCTGGCTCAG 3′) (Invitrogen Spain, TFS^®^, Waltham, MA, USA) [20,21]. That which was amplified was purified with the Clean up^®^ Nucleo spin Gel and PCR Kit. DNA concentration and purity were measured using DropVue^®^ (General Electric, Boston, MA, USA). The sequencing of the PCR products was carried out in the Genomics department of Complutense University of Madrid following the method described by Sanger et al. [22] with a 3730xl DNA Analyzer^®^ sequencer (Applied Biosystems, Foster City, CA, USA). The sequence was analyzed with the CLC Sequence Viewer 8 program and the EMBL-EBI multiple alignment tool, Clustal Omega (University College Dublin). Finally, the consensus sequence (1200 bp) was taken to the NCBI BLAST tool, obtaining the similarity of our sample with the sequences available in the *GeneBank* database.

### 2.3. Determination of PGPR Capacity

Assays of PGPR activity were tested from the isolated pure cultures. To determine the capacity to produce auxins, such as indolacetic acid (IAA) in vitro, a colorimetric technique was performed with Van Urk Salkowski reagent using the Salkowski method [23]. The isolates were grown in LB broth (Lennox, Richardson, TX, USA) and incubated at 28 °C for 4 days. The broth was centrifuged after incubation. Supernatant was reserved and 1 mL was mixed with 2 mL of Salkowski′s reagent (2% 0.5 FeCl_3_ in 35% HClO_4_ solution) and kept in the dark. The optical density (OD) was recorded at 530 nm after 30 min and 120 min. Results were quantified in ppm (mg/mL).

To determine the ACC degradation capacity of the bacterial strains, the protocol described by Glick [24] was followed, which also makes it possible to differentiate ACC-degrading bacteria from nitrogen-fixing (FN) bacteria. Plates were prepared with 1.8% Bacto-Agar (Difco Laboratories, Detroit, MI, USA), which has low nitrogen content, and spread with ACC (30 mmol). ACC must be fully dry before inoculation. Once inoculated, plates were incubated at 28 °C for 3 days and the growth was checked daily. The results were evaluated quantitatively (presence/absence).

The production of siderophores was detected using the Chrome Azurol S (CAS) agar medium described by Alexander & Zuberer [25]. The interpretation was based on the qualitative analysis of the production of a halo around the colonies, after 72 h of incubation at 28 °C, as the siderophore removes Fe from the Fe-CAS dye complex, which gives the medium its characteristic blue color.

The ability to solubilize inorganic phosphates was assayed following the protocol described by de Freitas et al. [26]. Tricalcium phosphate medium agar (TPM) [27] was used, adjusting final pH to 7 with 1 mol L^−1^ NaOH or HCl. After the inoculation and incubation for 72 h at 28 °C, well-separated colonies exhibiting distinguished clear zones (halos) were qualitatively evaluated (presence/absence) as solubilizers of inorganic phosphate.

### 2.4. Maximum Bactericidal Concentration of Hg (MBC)

For the study of the MBC of Hg of the selected strains, Müller Hinton agar plates of the commercial brand Pronadisa^®^ (Eucast, 2017, Växjö, Sweden) supplemented with HgCl_2_ at the following concentrations were seeded by exhaustion of the loop: 400, 350, 200, 175, 150, 100, 87.5, 75, 50, 43.75, and 25 µg/mL. The MBC was determined as the lowest concentration of HgCl_2_ capable of inhibiting the growth of more than 99.9% of the bacteria tested [19].

### 2.5. Bio-Mercury Remediation Suitability Index (BMRSI)

In order to comprehensively evaluate the strains that may be used for bioremediation purposes, the present work proposes the BMRSI obtained in accordance with the following formula:BMRSI = [IAA (mg/mL) + ACCd (1/0) + SID (cm) + PO43- (1/0)] + [MBC Hg (mg/mL)](1)
where: Presence = 1; Absence = 0.

### 2.6. Processing the Information

With the results of identification of the strains, calculations were made of the Shannon diversity index (H) and the modified maximum diversity (H_max_), in order to quantify the number of different types of microorganisms present in the sample (community). The protocol from Zak et al. was followed [28].
H = −∑p_i_ log_2_ p_i_(2)
where p_i_ is the proportion of characters belonging to the *i*th type in the community.

## 3. Results

### 3.1. Selection of Mercury-Tolerant Strains and Assessment of PGPR Activity

A total of 149 strains were isolated, of which 97 were Gram-positive, 44 Gram-negative, and eight were difficult-to-classify pleomorphic strains, from the different rhizospheres studied and from the bulk soil.

Given the interest in studying strains with potential biotechnological use, a first selective criterion was applied, according to which isolates showing at least one PGPR activity were included. Among all of the PGPR activities, the production of IAA was considered essential, being a positive result that exceeded 4 µg/mL. Using this criterion, 110 strains were preselected, and all of them had the following additional characteristics: (i) they are easily cultivable bacteria; and (ii) they present levels of Hg tolerance (CMB) higher than 50 µg/mL.

Next, the taxonomic identification of the 110 preselected strains was carried out, as well as the calculation of the Shannon-Wiener diversity index (H′) (Figure 1). The diversity of results fits the general pattern of frequency distribution in the sense that it is positively correlated with diversity and evenness [29]; the sampled community reveals few abundant and many rare species. Of the total of the 110 isolates, 46 belong to the genus *Bacillus*, (42% of the total). A total of 28 isolates belong to the genus *Pseudomonas* (25% of the total). 6% of the isolates could not be identified or did not have a high degree of parity by comparing their identification sequences in the Gene Bank, opening the possibility of the description of new species. These bacteria were classified as “undetermined bacterial species” (ND).

The analysis of the combined PGPR capacities of the different strains are shown in Figure 2. In said graph, the total number of strains that simultaneously present more than one PGPR activity have been counted, taking into account that the production of IAA is positive when it is above 4 µg/mL.

### 3.2. Maximum Bactericidal Concentration to Hg (MBC)

The highest MBC values obtained (350 µg/mL of HgCl_2_) are represented by strains 104, 114, 69-II, and 76, followed by a set of nine strains with values of 200 µg/mL. Most of the strains tolerated concentrations of around 100 µg/mL of HgCl_2_.

### 3.3. Bio-Mercury Remediation Suitability Index (BMRSI)

From the results obtained by the PGPR and CMB activities, the BMRSI was calculated. The results are aggregated in Table 1.

Based on the results obtained and in order to facilitate the selection of the most suitable strains due to their biotechnological relevance for subsequent biological tests, the following non-exclusive typologies were defined:Typology I: Strains with values equal to or greater than 6.5 in the BMRSI.Typology II: IAA producing strains above 5.5 µg/mL.Typology III: Strains producing three or more PGPR activities simultaneously.

Applying these criteria, a total of 39 strains appear in at least one of the three categories described (Table 2). The remaining 71 strains are relegated to second options for subsequent applied studies.

## 4. Discussion

The rhizosphere is a habitat that allows for the growth of a very abundant number of microorganisms. Just as the roots favor microbial growth, the development of the latter is also very important for the plant. Among these microorganisms, PGPRs have been successful in aiding plant growth, and those that could favor their phytoremediation capacity are especially interesting for the present study [30,31].

Success in remediation activity results from the joint activity of the microorganism and its interaction with plant species adapted to the type of hostile environment of contamination by heavy metals. Most of the work oriented toward the search for metallotolerant PGPR bacteria are usually oriented toward specific bacterial genera, such as *Bacillus* [32], *Azotobacter* [33], or *Pseudomonas* [34], among others. Additionally, in almost all cases the origin of isolation is that of plants for agricultural use [35,36,37], maximizing the probability of selection of strains with potential biotechnological use due to the selective pressure of the heavy metal and the coevolution/coadaptation with the plant.

In the case of Hg, several authors have demonstrated the existence of bacterial populations in the rhizosphere of various plants that have the capacity to tolerate and/or eliminate this heavy metal from the edaphic environment [38]. Other studies, such as those conducted by Nonnoi et al. [39] and Bekuzarova et al. [40], have isolated bacteria from root nodules of different plant species (*Medicago* spp. and *Trifolium* spp.), which grow in soils contaminated by Hg, even at high concentrations, and can be used for the removal of this heavy metal. The Hg resistance system is based on genes clustered in the mer operon that encode polypeptides with regulatory, transport, and enzymatic functions. They vary in number and identity of the genes that compose them and can have chromosomal characterization or be included in mobile gene elements such as plasmids and/or transposons. All of the mer genes of Gram positive bacteria are transcribed from a single promoter (merR (regulatory), merA (reductase), and merB (lyase), both transporter proteins). In Gram negative bacteria, merR is transcribed in the opposite direction to the transporter genes. The expression of the mer operon is inducible and is regulated by the product of the merR gene. When there is Hg^2+^ in the medium, the MerR-merO/P complex undergoes a conformational change allowing the RNA polymerase to transcribe the structural genes that make up the operon [41].

Given that mercury-tolerant and mercury-resistant microorganisms can contribute to reducing or eliminating the contamination of different forms of mercury in the environment, there is a growing interest in the selection of strains with potential biotechnological and bioremediative use [16].

A comprehensive analysis of the main activities of the PGPR that would allow for the selection of suitable strains for biotechnological uses has not been described in the scientific literature. Therefore, the present work proposes BMRSI, which harmoniously weights the importance of each variable in the equation, taking into account values published in previous studies for these equations. In this way, the comprehensive characterization of candidate strains for future germination, root formation, plant production, and edaphic colonization tests is facilitated. Comparison of the potential biotechnological use of the strains studied is simplified by obtaining a numerical value that quantifies the integral behavior of the strain, provided that the contaminant tolerance requirement (CMB) is met, and they have at least one activity, as PGPR favors plant recovery from the contaminated environment (IAA production).

Of all the PGPR activities, the production of auxins is especially important. Comparing our results with those obtained by other authors, we have found similar production values in Gram-positive strains, such as *Bacillus* spp. with similar biotechnological potential [42]. The concentrations of IAA produced by Gram-negative isolates are similar to those reported in the literature [43], or slightly lower on average than those found by other authors in similar studies with *Pseudomonas* spp. Against heavy metals, Gram-negative N-fixing and IAA-producing bacteria of the genera *Agrobacterium*, *Rhizobium*, *Klebsiella* and *Azotobacter* show productions between 4.90 μg/mL and 5.23 μg/mL, or slightly higher.

Siderophores are synthesized mainly by Gram-negatives, although cases of the genus *Bacillus* having produced these molecules have also been described [34]. In fact, radical exudates, particularly phenolic compounds, have a significant effect on the proliferation of siderophore-producing microorganisms in the rhizosphere of plants [44,45]. Bacteria such as *Bacillus subtilis* directly activate the iron acquisition strategy in plants. This mechanism can be explained by the production of volatile organic acids produced by the microorganism. Studies such as those carried out by Baldi et al., or Lewis et al. [46,47] show the importance of siderophores in the balance of metal ions in the medium, protecting bacteria from the hyper-accumulation of toxic metals. In the environment sampled for this study, the Hg concentration in the soil exceeds 7 ppm [48], a fact that could interfere with the production of some siderophores. For this reason, the importance of studying the production of this type of molecule in the presence of Hg has been considered in order to get as close as possible to future field test conditions.

Some soil bacteria promote the improvement of plant growth by reducing ethylene levels, which is linked to abiotic stress. Bacteria act in this process through the action of the enzyme ACC deaminase (ACCd), which deaminates the immediate precursor of ethylene, 1-aminocyclopropane-1-carboxylic acid (ACC) [49]. Plant improvement, and consequently growth, can be stimulated not only by a decrease in the ethylene content, but also by the generation of ammonium produced from ACC [49,50]. The levels of the ACCd enzyme vary widely in microorganisms, so plants positively select PGPR strains with this capacity in their rhizospheres in order to facilitate their growth and development. Therefore, it is not surprising that in the present study, 77% of the isolates that produce ACC came from rhizospheric soil.

Phosphate solubilizing microorganisms, which are recognized promoters of plant growth, use different mechanisms, such as the production of organic acids, which solubilize these insoluble phosphates in the rhizospheric zone. The phosphate solubilizing strains isolated in this study come entirely from the rhizosphere of the plants studied, which is associated with the possible recruitment of bacteria that improve and promote their growth and productivity by having access to bioavailable forms of phosphorus [51,52].

The study carried out by Ortiz Castro et al. [53], and more recently the work of Emami et al. [54], have concluded that the successful promotion of growth must be linked to the variety of bacterial mechanisms and activities which, in the environment of the plant′s rhizosphere, act synergistically during plant development. According to these researchers, and similar to what we have observed in the present study through the characterization of the strains using the BMRSI, the presentation of a single PGPR activity such as the synthesis of IAA does not guarantee the greatest success of a strain. It is the combination of several of the growth promoting activities, even in average values, that have a synergistic effect that improves plant development in a more noticeable manner. In this way, the evaluation of the bioremediation potential of the strain is facilitated, not only because of its effectiveness in favoring plant growth, but also because of its ability to do so in the unfavorable conditions that motivate the remediation action, such as the presence of a heavy metal like Hg. Based on the foregoing, and in view of the results obtained for the group of strains studied, a BMRSI value greater than or equal to 6.5 has been set as the cut-off point for a strain to be considered of interest for subsequent biological tests.

The *Bacillus* and *Pseudomonas* genera are especially abundant in the composition of the soil microbiota in numerous studies. Of all the species isolated in the present study, *Bacillus toyonenesis* had the highest representation among those that are Gram-positive, all of which were identified with a percentage of similarity greater than or equal to 99%. This microorganism was described by Jiménez et al. [55] as a new species within the *Bacillus cereus* group. This microorganism has been used as a probiotic in livestock feed. There are recent references that describe the use of this bacterium as PGPR thanks to its remarkable production of IAA [56]. Similarly, their tolerance to Hg has been demonstrated as described by Naguib et al. [35]. In view of the existing references and the high BMRSI values that this species shows in the present study (with the exception of strain 22, all of the values are systematically greater than 6.5), it is considered to be a good candidate for continued evaluation of its suitability for future use in bioremediation.

*Brevibacterium frigoritolerans* and *Bacillus drentensis* were isolated with the same abundance among Gram-positives and both have been identified with a similarity index greater than 99%. Tolerance to Hg has been described in *Brevibacterium frigoritolerans* [57]. In turn, Khezrinejad et al. [58] propose the use of this bacterium as PGPR due to the fact that it is a strong producer of IAA, with levels of production of these phytohormones similar to those found in the present study. *Bacillus drentensis* has been used since 2016 by research groups such as Mahmood et al. [59], and more recently Arikan & Pirlak [60], in conjunction with *Enterobacter cloacae* to improve tolerance to salinity of various plant species, having obtained very good results from the experiments, as there are also studies that support the correlation between resistance to heavy metals in general, including Hg, and tolerance to salinity [61].

Several authors have demonstrated the potential of *Bacillus aryhabattai* as a bacteria capable of promoting growth in many plant species. Lee et al. [62] verified its effects on *Xanthium italicum*, obtaining good results and proposing it as an interesting strain that could be used to revegetate sterile soils. Along the same lines, Park et al. [63] proved that this bacterium is a valuable resource for use in biofertilizers and other soil amendments that seek to improve crop productivity, because of the significant production of phytohormones (abscisic acid, indole acetic acid, cytokinin, and different gibberellic acids). The isolates of this strain in the present study also stand out for the production of IAA above 5 μg/mL.

Isolate 217 presents a 98% BLAST homology index with the *Bacillus nealsonii* species, and consequently, by applying the sequence homology criteria with the database available in GeneBank, we can only assume that it belongs to the *Bacillus* genus. In the literature, however, there are no references that refer to the possible PGPR effect of this species. Therefore, it would be advisable to determine the identity of this isolate in order to verify that it belongs to this species.

The prevalent genus of Gram-negative bacteria is *Pseudomonas*, with 41% of all identified microorganisms. Five isolates correspond to the *phaseolicola* pathovariety, so despite the fact that some of the strains studied have good PGPR potential, their possible biotechnological use in bioremediation should be carefully evaluated, since many strains of this species are phytopathogenic [64].

Four isolates have been identified as *Pseudomonas brasicacearum*, of which three belong to the subspecies brasicacearum (*P. brasicacearum* subsp. *brasicacearum*) and one to the subspecies neoaurantiaca (*P. brasicacearum* subsp. *neoaurantiaca*). This species was already proposed by Zachow et al., [65] along with other species of the genus *Pseudomonas* (among which it is worth highlighting *Pseudomonas corrugata*, which we will refer to later), as an interesting strain for commercial use in the promotion of agricultural plants, helping to make crops more tolerant to different types of stress. In the present study, the isolates of this species are postulated as good candidates for further use in bioremediation, as they present high BMRSI (up to 7.4) and high tolerance to Hg (up to 200 μg/mL). As we mentioned previously, *Pseudomonas corrugata* has also been used with good results as a suppressor rhizobacteria for fungal diseases and biocontrol [66,67,68]. Three of the isolated strains of this species reached the maximum resistance to Hg measured in the present study, with strain 69-II tolerating concentrations of up to 350 μg/mL.

The use of *Pseudomonas fluorescens* as PGPR has already been tested by various authors. Among them are Gamez et al. [69], who observed significantly higher improvement in banana plants (*Musa acuminata*) when inoculating these bacteria in its rhizosphere. Similarly, other studies confirm the use of strains of this species in the plant recovery of soils with high levels of salinity [70]. Moreover, Sivasakthi et al. [71] inoculated strains of these species in crops to promote plant growth in field trials, obtaining improvements in yield of up to 44% thanks to their rapid root colonization capacity. Regarding the ability of this species to tolerate high concentrations of Hg, various authors have studied and described the ability of *Pseudomonas fluorescens* to resist various concentrations of Hg, as well as its potential use in bioremediation by having the ability to biosorb and biotransform this heavy metal [72,73].

Finally, there are four unidentified strains (due to their low degree of parity in contrast to the GeneBank data) which, due to their high BMRSI values, should be studied in depth in future trials.

## 5. Conclusions

In view of the results obtained, in which the bioremediation potential of strains of a diverse nature has been evaluated, the BMRSI can be considered a tool that allows for the harmonious, global weighting of the behavior of the different PGPR activities, as well as for discriminating and selecting those strains with greater probability of success in soil recovery through phytoremediation, especially when carried out in environments with the external pressure of a pollutant such as Hg.

## Figures and Tables

**Figure 1 ijerph-18-04213-f001:**
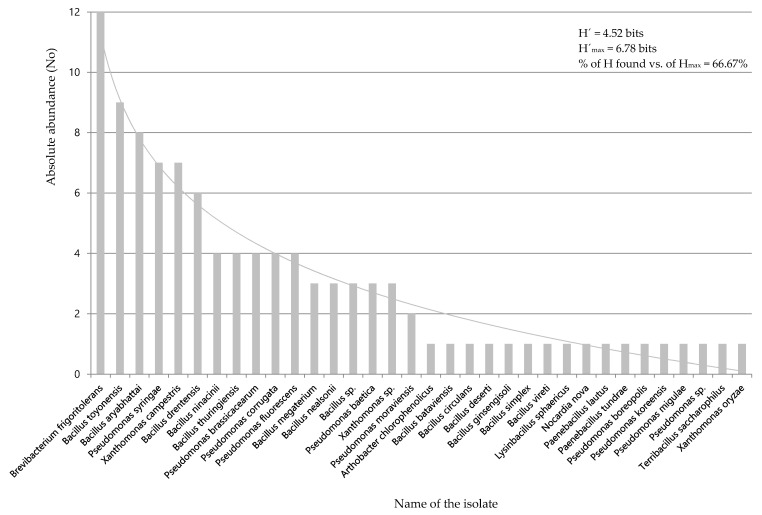
Sorting by absolute abundances of the isolated strains. The curve represents the trend found. Strains “Not determined” (ND) are not included. The figure shows the values of H, H_max_ and the percentage of Hmax found. −∑p_i_ log_2_ p_i_.

**Figure 2 ijerph-18-04213-f002:**
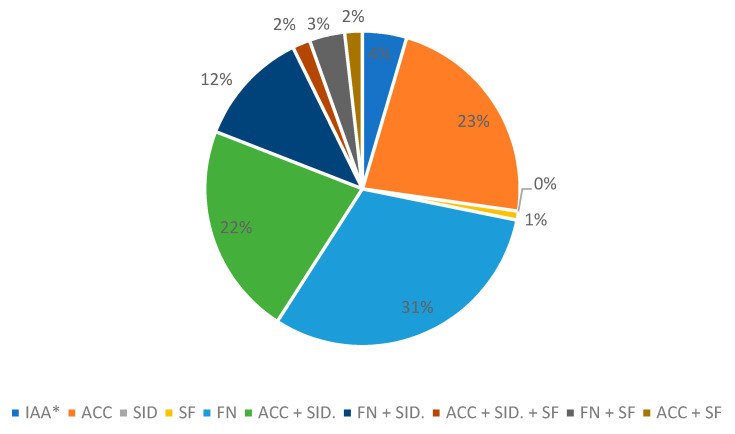
Description of the plant growth promoting rhizobacteria (PGPR) activities of the isolates. IAA: production of auxins (indolacetic acid); ACC: production of 1-aminocyclopropane-1-carboxylic acid (ACC) deaminase; SID: production of siderophores; SF: solubilization of phosphates, FN: nitrogen fixers, (*): strains that have a single PGPR activity (production of auxins).

**Table 1 ijerph-18-04213-t001:** Bio-Mercury Remediation Suitability Index for the tested strains.

**Strain**	**69-II**	**80**	**74**	**130**	**146**	**25**	**18**	**69-I**	**211**	**212**	**11**	**43**	**95**	**20**	**79**
BMRSI	8.51	8.42	8.07	8.01	7.99	7.89	7.87	7.85	7.74	7.73	7.69	7.68	7.57	7.55	7.55
**Strain**	**10**	**31**	**57**	**55**	**21**	**50**	**175**	**37**	**98**	**76**	**23**	**204**	**1**	**48**	**173**
BMRSI	7.42	7.4	7.26	7.23	7.21	7.08	7.08	7.07	7.05	7.04	6.97	6.8	6.68	6.62	6.6
**Strain**	**122**	**9**	**58**	**56**	**159**	**70**	**214**	**114**	**160**	**75**	**149**	**186**	**35**	**168**	**166**
BMRSI	6.59	6.56	6.46	6.43	6.38	6.35	6.34	6.32	6.32	6.3	6.26	6.23	6.21	6.09	6.03
**Strain**	**178**	**167**	**217**	**104**	**26**	**133**	**213**	**19**	**22**	**118**	**121**	**151**	**155**	**112**	**161**
BMRSI	6.00	5.93	5.93	5.86	5.84	5.83	5.82	5.81	5.75	5.71	5.69	5.63	5.61	5.61	5.6
**Strain**	**47**	**14**	**16**	**154**	**200**	**88**	**223**	**203**	**174**	**190**	**199**	**206**	**195**	**126**	**68**
BMRSI	5.58	5.51	5.47	5.46	5.46	5.41	5.35	5.34	5.33	5.33	5.32	5.31	5.3	5.29	5.25
**Strain**	**224**	**30**	**189**	**128**	**162**	**137**	**117**	**216**	**5**	**197**	**191**	**196**	**109**	**180**	**192**
BMRSI	5.23	5.23	5.2	5.2	5.2	5.17	5.16	5.15	5.11	5.05	5.00	4.94	4.91	4.9	4.86
**Strain**	**201**	**124**	**134**	**45**	**106**	**135**	**96**	**108**	**142**	**145**	**82**	**153**	**91**	**143**	**210**
BMRSI	4.82	4.79	4.79	4.77	4.76	4.75	4.73	4.71	4.69	4.55	4.53	4.52	4.47	4.44	4.39
**Strain**	**125**	**132**	**139**	**188**	**4**										
BMRSI	4.34	4.34	4.32	4.3	4.26										

**Table 2 ijerph-18-04213-t002:** List of the thirty-nine strains selected in the second screen based on their PGPR activity. No.: strain number, SL: bulk soil, A: *Rumex induratus*, B: *Rumex bucephalophorus*, C: *Avena sativa*, D: *Medicago sativa*, E: *Vicia bengalensis*. BMRSI: Bio-Mercury Remediation Suitability index; “ND” not described strain.

No.	RF/SL	MBC (µg/mL)	BMRSI	IAA (µg/mL)	ACCd (p/a)	SID. (cm)	SOL. PO_4_^3−^	IDENTIFICATION
**1**	SL	50	6.68	4.63	-	1	-	*Bacillus toyonensis*
**9**	SL	75	6.56	5.59	+	-	-	*Bacillus toyonensis*
**10**	SL	200	7.42	6.12	-	1.1	-	ND
**11**	SL	87.5	7.69	5.61	-	1	-	*Bacillus toyonensis*
**18**	SL	100	7.87	6.28	+	0.5	-	*Bacillus toyonensis*
**20**	SL	100	7.55	5.96	+	0.5	-	*Bacillus toyonensis*
**21**	SL	100	7.21	5.31	+	0.8	-	*Bacillus toyonensis*
**22**	SL	87.5	5.75	4.57	+	0.1	-	*Bacillus toyonensis*
**23**	SL	175	6.97	4.89	+	0.9	-	*Pseudomonas moraviensis*
**25**	SL	150	7.89	5.85	+	0.9	-	*Bacillus toyonensis*
**31**	A	100	7.4	5.6	+	0.7	-	*Pseudomonas brassicacearum subsp. brassicacearum*
**37**	A	87.5	7.07	5.58	-	0.5	-	*Bacillus aryabhattai*
**43**	A	87.5	7.68	5.7	+	0.9	-	*Bacillus toyonensis*
**48**	A	100	6.62	4.92	+	0.6	-	ND
**50**	A	100	7.08	5.29	+	0.7	-	*Bacillus toyonensis*
**55**	A	87.5	7.23	5.56	-	0.8	-	*Pseudomonas brassicacearum sbups. neoaurantiaca*
**56**	B	200	6.43	4.43	+	0.8	-	*Pseudomonas brassicacearum subsp. brassicacearum*
**57**	B	175	7.26	6.38	+	0.6	-	*Pseudomonas syringae pv. phaseolicola*
**58**	B	100	6.46	5.56	+	0.7	-	*Pseudomonas brassicacearum subsp. brassicacearum*
**69-I**	B	75	7.85	6.08	-	0.7	-	*Pseudomonas corrugata*
**69-II**	B	350	8.51	5.71	+	0.7	+	*Pseudomonas corrugata*
**74**	B	100	8.07	6.27	+	0.7	-	*Pseudomonas syringae pv. phaseolicola*
**76**	B	350	7.04	4.99	+	0.7	-	*Pseudomonas syringae pv. phaseolicola*
**79**	B	87.5	7.55	5.27	+	0.4	-	*Pseudomonas syringae pv. phaseolicola*
**80**	B	80	8.42	6.47	+	0.8	-	*Pseudomonas syringae pv. phaseolicola*
**95**	C	80	7.57	4.69	-	2.8	-	*Brevibacterium frigoritolerans*
**98**	C	160	7.05	5.29	+	0.6	-	*Pseudomonas baetica*
**112**	C	150	5.61	4.36	+	0.1	-	*Pseudomonas corrugata*
**122**	D	87.5	6.59	4.51	+	-	+	*Brevibacterium frigoritolerans*
**130**	D	160	8.01	5.85	+	1	-	*Pseudomonas corrugata*
**146**	E	80	7.99	6.09	+	0.8	-	*Pseudomonas fluorescens*
**151**	E	87.5	5.63	4.38	+	0.2	-	*Bacillus aryabhattai*
**168**	A	87.5	6.09	4.00	+	-	+	*Bacillus aryabhattai*
**173**	A	175	6.6	5.53	+	-	-	*Bacillus toyonensis*
**175**	A	80	7.08	6.00	+	-	-	ND
**204**	D	80	6.8	5.72	-	-	+	ND
**211**	D	80	7.74	6.16	+	0.5	-	*Bacillus drentensis*
**212**	D	80	7.73	6.16	+	0.4	-	*Bacillus drentensis*
**217**	E	100	5.93	4.88	+	2	+	*Bacillus nealsonii*

## Data Availability

No new data were created or analyzed in this study. Data sharing is not applicable to this article.

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
