# Peer review of "Bio-Mercury Remediation Suitability Index: A Novel Proposal That Compiles the PGPR Features of Bacterial Strains and Its Potential Use in Phytoremediation"

_ijerph, 2021, doi:10.3390/ijerph18084213_

Round 1

Reviewer 1 Report

In this manuscript, the authors propose a Bio-Mercury Remediation Suitability Index (BMRSI) for screening potential strains for contamination remediation. Overall the authors have done very diligent experiments and the manuscript is somewhat interesting. I have the following main concerns:
1, the main microorganisms screened were in two genera (bacillus & Pseudomonas), and although the authors present some information about these two genera in their discussion, the promotion of plant growth in the under heavy metals and the mechanisms of heavy metal detoxification by these two genera are not adequately described
2, In practical applications, the role of microorganisms is influenced by more complex and thus in culture conditions. It is rather unfortunate that the authors did not conduct pot experiments. At least these potential effects should have been described in the discussion section.
3, microorganisms play a more complex role under Hg contamination, like regulating the Hg cycle. There is a lot of work illustrating these and the authors should have taken them into account.

Author Response

First of all, we want to thank the time and dedication, as well as the interesting contributions made. It is always pleasant to check that the article has been carefully reviewed and that the contributions help to reflect and develop the research.

  1. The main microorganisms screened were in two genera (bacillus & Pseudomonas), and although the authors present some information about these two genera in their discussion, the promotion of plant growth in the under heavy metals and the mechanisms of heavy metal detoxification by these two genera are not adequately described.

In order to solve this point, a description has been included (lines 253-268, highlighted in yellow) of the main mechanism that bacteria have to detoxify the environment from the presence of Hg, through the expression of genes contained in the mer operon, which looks as follows:

"The Hg resistance system is based on genes clustered in the mer operon that encode polypeptides with regulatory, transport, and enzymatic functions. They vary in number and identity of the genes that compose them and can have chromosomal characterization or be included in mobile gene elements such as plasmids and / or transposons. All the mer genes of Gram positive bacteria are transcribed from a single promoter (merR (regulatory), merA (reductase) and merB (lyase), both transporter proteins). In Gram negative bacteria, merR is transcribed in the opposite direction to the transporter genes. The expression of the mer operon is inducible and is regulated by the product of the merR gene. When there is Hg2 + in the medium, the MerR-merO / P complex undergoes a conformational change allowing the RNA polymerase to transcribe the structural genes that make up the operon [41].

Given that mercury-tolerant and mercury-resistant microorganisms can contribute to reducing or eliminating the contamination of different forms of mercury in the environment, there is a growing interest in the selection of strains with potential biotechnological and bioremediative use [42]".

Likewise, the main differences in its regulation present in Gram negative and Gram positive ones are presented. Enter into detailing the specific mechanisms of Pseudomonas spp and Bacillus spp. would take a long development and we understand that the focus of this paper is more oriented to the practical aspect of the application of the BMRSI than to the molecular aspects of resistance regulation. Please assess the adequacy of its inclusion in the main text. All this information has been collected in a very exhaustive and synthetic way by Boyd et al., (2012) in their article “The mercury resistance operon: from an origin in a geothermal environment to an efficient detoxification machine”.

  1. In practical applications, the role of microorganisms is influenced by more complex and thus in culture conditions. It is rather unfortunate that the authors did not conduct pot experiments. At least these potential effects should have been described in the discussion section.

In accordance with the information provided by the reviewer, we are aware of the need to consider as many variables as possible to recreate, in the most faithful way possible, the reality under study.

The influence that different external and more complex factors (such as the presence of Hg) may have on the ability to promote microbial growth by the strains tested is well characterized. In this sense, we have carried out different trials in which we have studied the impact of Hg in original soils of the Almadén mining district on plant physiology (biometric and enzymatic parameters related to oxidative stress) in the presence of PGPR strains with good results and that we hope to publish soon. These results have not been included in this article, focused from the perspective of microbial ecology, as they are considered to exceed the scope, focused on proposing and verifying the employability and adequacy of the BMRSI, rather than physiological aspects of the PGPR capacity of the strains studied. However, below, we attach an aggregate table with some of the data that we have obtained and that are part of a study that we are developing (unpublished results) but that, in any case, respond to the reviewer's considerations (Table 1).

SUBSTRATE

TREATMENT

(AVERAGE AGGREGATE)

DRY WEIGHT OF THE AERIAL PART (g)

LENGTH OF THE AERIAL PART (cm)

Soil in presence of Hg (Plot 6)

CONTROL

3,33

12,70

INOCULUM WITH THE STRAIN

3,49

12,84

INOCULUM WITH CONSORTIUM

3,72

13,82

Soil without Hg (Plot 9)

CONTROL

2,17

10,28

INOCULUM WITH THE STRAIN

2,74

14,77

INOCULUM WITH CONSORTIUM

2,79

13,99

Table 1. Influence of the presence of Hg and type of treatment of the seeds of L. albus var. Marta on the biometric parameters (i) dry weight of the aerial part (g) and (ii) length of the aerial part (cm). Aggregated data whose origin is an article in development and that will be published soon, with a greater level of detail and analysis.

In Table 1 what is observed, very briefly, is the positive effect of the strains and their consortia with respect to their respective controls (presence and absence of Hg), in the biometric parameter "dry weight of the aerial part”. Likewise, it is observed that, despite the negative effect of the presence of Hg in the medium on the length of the aerial part, the presence of strains and their consortia seem to favor a greater development of this biometric parameter.

These tests have been carried out in the presence of soil contaminated with Hg collected from the mining district of Almadén (Plot 6; 1720 mgHg / kg, Millán et al., 2007) (described in the text as one of the environments with the highest levels of contamination worldwide by this heavy metal). Homology with soils in low Hg concentration is carried out in the same mining district (in order to preserve the physical / chemical and edaphic cross-sectional homogeneity of the samples) at low Hg concentrations (Plot 9; 122.4 mgHg / kg, Millán et al., 2007). It is important to emphasize that these data serve as sample data since they are part of a larger study with promising results, which will be published soon and which will focus on the impact of Hg on plant physiology (biometric level and enzymatic response to oxidative stress), as well as the role of the best candidate strains on plant growth under these conditions.

  1. Microorganisms play a more complex role under Hg contamination, like regulating the Hg cycle. There is a lot of work illustrating these and the authors should have taken them into account.

Indeed, we agree with this contribution. In fact, it is well described that microorganisms exert part of their beneficial effect on the plant by interfering in the mobilization of Hg from the edaphic environment, which is considered an indirect way of promoting plant growth. However, in this article we focus on the direct aspects that the strains exert on the plant through the main PGPR activities (Restrepo-Franco et al., 2015; Patten and Glick, 2002; Spaepen et al., 2007; Glick , 2010; Ullah et al., 2015).

What the reviewer proposes is part of a later trial (under development), of which we already have preliminary results, in which we verify the phytorhizoremediation capacity of Hg in the presence of the best candidate strains (BMRSI> 6) and which exert a favorable response in the growth of the plant grown in soils contaminated with Hg (in vivo) (Plot 6; 1720 mgHg / kg, Millán et al., 2007). The objective of this test is to verify the mobilizing capacity of Hg in the presence of the selected PGPR strains and to verify the different bioaccumulation and / or volatilization of Hg under the in vivo conditions tested.

Reviewer 2 Report

Comments

(Manuscript No. ijerph-1148566)

The manuscript titled “Bio-Mercury Remediation Suitability Index: A Novel Proposal that Compiles the PGPR Features of Bacterial Strains and its Potential Use in Phytoremediation” is well written and presents Bio-Mercury Remediation Suitability Index (BMRSI) for a comprehensive assessment of strains that may be used for bioremediation purposes. The importance of this manuscript is that its results can provide theoretical support for future research on biological methods to solve mercury pollution. Overall, the mechanism reported in this manuscript is very important. I strongly recommend publishing this manuscript with minor revisions:

(1) It is recommended to change the "ml" in the article to "mL".

(2) The format of the numbers in Figure 1 is wrong.

(3) There is a format error in the title of Figure 1.

(4) It is recommended to remove the black border in Figure 2.

Author Response

First of all, we want to thank the time and dedication, as well as the interesting contributions made. It means a lot to know that the article is valued in detail and that the proposals made represent an improvement.

Below you will find the response to your contributions:

  1. It is recommended to change the "ml" in the article to "mL". Changes marked in yellow in the text.
  2. The format of the numbers in Figure 1 is wrong. Changes marked in yellow in the text.
  3. There is a format error in the title of Figure 1. Changes marked in yellow in the text.
  4. It is recommended to remove the black border in Figure 2. The black border has been removed.

Thank you

Reviewer 3 Report

This paper reads well and adds to the body of knowledge.  However, the following changes are required before the publication:

  1. The title of the paper includes 'Phytoremediation', however, the content of the paper is focusing on 'bioremediation'.  This need to be clarified.
  2. Line 38: remove "so-called"
  3. Line 155: what is the term "free Soil" refers to? use other terms.
  4. Figure 1: include the unit as well as the title on the Y-axis.  Also increase the font on X-axis.  provide title for the X-axis
  5. Line 177: change 'Table' to 'Figure'
  6. Provide more detail about 'Table 2'
  7. The introduction needs to be extended to include more background about the mercury (HG):

    • what are the sources of HG in the environment?
    • How Hg is toxic? what are some of the health effect on human, etc.?
    • How the HG can get into the food chain? provide some explanation.

Author Response

First of all, we want to thank the time and dedication, as well as the interesting contributions made. It means a lot to know that the article is valued in detail and that the proposals made represent an improvement.

Below you will find the response to your contributions: 

  • The title of the paper includes 'Phytoremediation', however, the content of the paper is focusing on 'bioremediation'. This need to be clarified.

This is a very interesting contribution that we were evaluating before submitting the article. Finally, we understand that bioremedation is a concept that includes many practical in situ and ex situ approaches (Azubuike et al., 2016). The focus of this work is to generate a tool for the search for bacteria that, due to their tolerance to Hg (or potentially other heavy metals) and their PGPR capacity, are coadjuvants of an optimal physiological state of plants that can clean in situ. heavy metal. This, in a strict sense, is a process of Phytoremediation (more specifically phyto-rhizo-remediation). It is only for this ultimate goal (using PGPRs in the plant rhizosphere for remediation) that we use that term. It would not make sense, if not for this last purpose, to screen the strains to collect good PGPR.

  • Line 38: remove "so-called". Changes marked in yellow on the text.
  • Line 155: what is the term "free Soil" refers to? use other terms. Changes marked in yellow on the text. Free soil was unproperly translated and it’s been replaced with “bulk soil”.
  • Figure 1: include the unit as well as the title on the Y-axis. Also increase the font on X-axis.  provide title for the X-axis. Changes included in the table.
  • Line 177: change 'Table' to 'Figure'. Changes marked in yellow in the text.
  • Provide more detail about 'Table 2'
  • The introduction needs to be extended to include more background about the mercury (HG):
    1. What are the sources of HG in the environment? Changes marked in yellow in the text. Lines 38-49
    2. How Hg is toxic? what are some of the health effect on human, etc.? Changes marked in yellow in the text. Lines 33-35
    3. How the HG can get into the food chain? provide some explanation. Changes marked in yellow in the text. Lines 30-33

Thank you,

M

Round 2

Reviewer 1 Report

I am satisfied with the changes.